# Chemical Composition and Nutritional Value of Different Species of *Vespa* Hornets

**DOI:** 10.3390/foods10020418

**Published:** 2021-02-14

**Authors:** Sampat Ghosh, Saeed Mahamadzade Namin, Victor Benno Meyer-Rochow, Chuleui Jung

**Affiliations:** 1Agriculture Science and Technology Research Institution, Andong National University, Andong 36729, Korea; sampatghosh.bee@gmail.com (S.G.); saeedmn2005@gmail.com (S.M.N.); meyrow@gmail.com (V.B.M.-R.); 2Department of Plant Protection, College of Agriculture, Varamin-Pishva Branch, Islamic Azad University, Varamin 3381774895, Iran; 3Department of Ecology and Genetics, Oulu University, 90140 Oulu, Finland; 4Department of Plant Medicals, Andong National University, Andong 36729, Korea

**Keywords:** *Vespa velutina*, *Vespa mandarinia*, *Vespa basalis*, entomophagy, amino acids, fatty acids, minerals

## Abstract

We genetically identified three different species of hornets and analyzed the nutrient compositions of their edible brood. Samples were collected from a commercial production unit in Shizong province of China and from forests near Andong City in Korea. The species were identified as *Vespa velutina*, *V. mandarinia*, and *V. basalis* from China and *V. velutina* from Korea. Farmed *V. velutina* and *V. mandarinia* were found to have similar protein contents, i.e., total amino acids, whereas *V. basalis* contained less protein. The *V. velutina* brood collected from the forest contained the highest amount of amino acids. Altogether 17 proteinogenic amino acids were detected and quantified with similar patterns of distribution in all three species: leucine followed by tyrosine and lysine being predominant among the essential and glutamic acid among the non-essential amino acids. A different pattern was found for fatty acids: The polyunsaturated fatty acid proportion was highest in *V. mandarinia* and *V. basalis*, but saturated fatty acids dominated in the case of *V. velutina* from two different sources. The high amounts of unsaturated fatty acids in the lipids of the hornets could be expected to exhibit nutritional benefits, including reducing cardiovascular disorders and inflammations. High minerals contents, especially micro minerals such as iron, zinc, and a high K/Na ratio in hornets could help mitigate mineral deficiencies among those of the population with inadequate nutrition.

## 1. Introduction

Numerous communities in the world traditionally include the broods of wasps and hornets (Hymenoptera) in their diets (Figure 1). Since ancient times it has been customary for Chinese people, especially those from Yunnan, to consume wasps, as is documented in a book from the Tang dynasty (618–907 C.E.) [1]. In Korea, *Vespa* hornet nests and larvae are harvested for medicinal and occasional edible use, especially *V. mandarinia* [2]. In Laos insects including wasps and hornets are consumed by 95% of the population, with the exception of two provinces for which a value of 85–90% has been reported [3]. The hornets consumed are of a different species, but *V. affinis* and *V. tropica* are particularly popular in Laos [4], whereas *V. cincta* (synonym to *V. tropica,* based on a report by [5]) is favored in Northern Thailand [6]. Wasp pupae as well as the adults belonging to the genus *Vespula,* locally known as *hebo*, are popular in the mountainous region of Honshu, Japan [7,8]. People commonly harvest the wasp nest from the forest, but in some cases the wasps are semi-domesticated [4,8] (Figure 1B).

The preparation of the wasp and hornet brood for consumption varies. The most common method of preparation includes deep-frying or frying with chicken eggs as in Yunnan, but the Dai people of southern Yunnan prefer to steam larvae and pupae together and mix them with vinegar and other seasonings [1]. Korean people also prefer frying and roasting [2], whereas Japanese people cook or preserve the wasps using soy sauce and mirin (sweet cooking rice wine) [8]. Although in Korea hornets are not considered a common food, they did, however, feature in the armamentarium of Korean traditional medicine. *Polistes* and *Vespa* larvae and adults are used as therapeutics for different ailments [9,10].

*Vespa* spp. have received attention not only as edible insects, but also as a pest affecting honeybee populations. In this context *V. velutina*, an invasive species, is regarded as the most obnoxious in Korea [11,12] and in Europe as well [13]. Although the Asian honeybee, *Apis cerana,* has evolved a “heat balling defense” and warning behavior when hornets are patrolling near their nests, the European honeybee, *A. mellifera*, does not possess this behavior and is therefore more susceptible to an attack by *V. velutina* [14,15,16,17]. A study of risk prediction in connection with the distribution of *V. velutina* [18,19] shows that the yearly dispersal of the species is 9.4 km northward in Korea and that this could have a serious impact on beekeeping as well as biodiversity and the ecosystem in general.

Nine native and one invasive hornet species occur in South Korea: *V. analis*, *V. binghami*, *V. mandarinia*, *V. simillima simillima*, *V. simillima xantothorax*, *V. crabro crobroformis*, *V. crabro flavofasciata*, *V. ducalis*, *V. dybowskii*, and *V. velutina nigrothorax* [20,21]. The lifecycle of the Vespa species includes different phases. Following overwintering, the emergence foundress, i.e., mated queen, searches for a suitable nest site and then constructs the primary nest for egg laying. About a month later, the eggs will have developed into female, non-reproductive adult workers and predation begins. During that time the workers build a secondary nest that is bigger in size than the primary nest. After that, large numbers of future queens are produced, which mate with drones that stem from unfertilized eggs. As the winter approaches, drones and workers die, while queens seek out overwintering places and the cycle continues the following spring [13].

Harvesting these wasps and hornets (*Vespa* spp.) can be a mode of biocontrol, but it can also lead to the use of these species as human food or animal feed. The present study was undertaken to assess the nutrient composition of three different species of *Vespa.* It was hoped that an investigation such as this would give us an opportunity to understand why there are differences in amino acid as well as fatty acid compositions between semi-domesticated hornets and specimens collected from the wild, e.g., the forest environment.

## 2. Materials and Methods

### 2.1. Sample Collection and Preparation

Three bottles of *Vespa* broods, representing three species, were obtained in frozen and dried form from a hornet-producing farm in Shizong County in China. The hornets were semi-domesticated primarily to be sold as food. All of the *Vespa* samples were packed into a freezing box and brought to the laboratory (Andong National University, Andong, South Korea). Samples were stored at −20 °C until further processing. A few individuals (*n* = 10) from each bottle were taken for our DNA barcoding experiment in order to identify the species. Specimens used in the chemical analyses were not separated according to developmental stage; they included late instar larvae and pupae together, almost 50:50 (as this is the way they are sold and used by the consumer), but excluded adults.

A *V. velutina* nest was harvested in early morning from the forest behind the university (Andong City, Korea) in the month of July and brought to the laboratory. The broods were collected, similar to the Chinese commercial late instar larvae and pupae in a 50:50 proportion, separately from the nest and kept in a refrigerator (−20 °C) until further processing (*n* = 100), which involved freeze drying and grinding up into a homogenous powder form.

### 2.2. Identification of the Species

The collected specimens i.e., broods of *Vespa*, were labelled VEUN20, VENU21, and VENU22 and identified based on DNA barcoding. The total DNA of each sample was extracted from the head and thorax using a DNeasy Blood & Tissue kit (QIAGEN, Inc., Dusseldorf, Germany) following the manufacturer’s protocol. Two primers, LCO-1490 (5′-GGT CAA CAA ATC ATA AAG ATA TTG G-3′) and HCO-2198 (5′-TAA ACT TCA GGG TGA CCA AAA AAT CA-3′) targeting mitochondrial the *Cytochrome Oxidase I* (COI) gene [22] were used. The polymerase chain reaction (PCR) was conducted using AccuPower PCR PreMix (Bioneer, Daejeon, Korea) in order to amplify the COI gene corresponding to “DNA Barcode” region [23]. Sequencing was performed commercially by Macrogen (Seoul, South Korea). All three sequences were generated in both directions and assembled using Bioedit v7.0.5.2 [24] to annotate the species level identification using the BLAST (Basic Local Alignment Search Tool) database of the National Center for Biotechnology Information (NCBI) (http://www.ncbi.nlm.nih.gov, accessed on 31 August 2019). Based on the similarity (in %) the specimens were identified.

### 2.3. Nutritional Composition Analyses

#### 2.3.1. Amino Acid Analysis

The amino acid composition was estimated using a Sykam Amino Acid analyzer S433 (Sykam GmbH, Eresing, Germany) following a standard method of AOAC (Association of Official Analytical Chemists) [25]. The *Vespa* samples in powder form were hydrolyzed in 6 N HCl for 24 h at 110 °C under a nitrogen atmosphere followed by concentrating in a rotary evaporator. The concentrated samples were reconstituted with sample dilution buffer provided by the manufacturer (0.12N citrate buffer, pH 2.20). The hydrolyzed samples were analyzed for amino acid composition. The amino acid score was calculated considering the total estimated amino acid as protein, based on the WHO/FAO/UNU (World Health Organization/Food and Agriculture Organization/United Nations University) [26] report of a joint WHO/FAO/UNU Expert Consultation on protein and amino acid requirements in human nutrition following the formula [27]:$$\mathrm{Amino}\;\mathrm{acid}\;\mathrm{score}=\frac{\left(\mathrm{mg}\;\mathrm{of}\;\mathrm{amino}\;\mathrm{acid}\;\mathrm{in}\;1g\;\mathrm{of}\;\mathrm{test}\;\mathrm{protein}\right)\;\times \;100}{\mathrm{mg}\;\mathrm{of}\;\mathrm{amino}\;\mathrm{acid}\;\mathrm{in}\;\mathrm{reference}\;\mathrm{pattern}}$$

#### 2.3.2. Fatty Acid Composition Analysis

Fatty acid compositions of studied *Vespa* were determined and quantified using gas chromatography–flame ionization detection (GC-14B, Shimadzu, Tokyo, Japan) equipped with an SP-2560 column, following the recommended method of the Korean Food Standard Codex [28]. Briefly, the samples were derivatized into fatty acid methyl esters (FAMEs), which were then identified and quantified by comparing the retention time and peak areas of standards from Sigma (Yongin, Korea) and analyzed under the same conditions.

#### 2.3.3. Mineral Analysis

Minerals of nutritional importance were analyzed following standard procedures of the Korean Food Standard Codex [28]. Dried *Vespa* powder samples were digested with nitric and hydrochloric acid (1:3) at 200 °C for 30 min in a high pressure microwave digestive system. The mineral contents, upon filtration with 0.45 micron filter paper, were analyzed using an inductively coupled plasma-optical emission spectrophotometer (ICP-OES 720 series; Agilent; Santa Clara, CA, USA). Recommended dietary allowance (RDA), population reference intake (PRI), and adequate intake (AI) values for the respective minerals were obtained from organizations such as the Linus Pauling Institute of Oregon State University and the European Food Safety Authority (EFSA).

#### 2.3.4. Statistical Analysis

Composite sampling methods were followed including 100 samples for each group. In order to increase reliability the chemical analysis was carried out in at least duplicate and represented as mean ± standard deviation. To test the differences for individual nutrients of different *Vespa*, we carried out one-way ANOVA (analysis of variance) followed by a post hoc test (Tukey’s Honestly Significant Difference (HSD)) using SPSS 16.0 (SPSS Inc., Chicago, IL, USA). If the *p*-value was found to be ≤0.05 (CI = 95%), the null hypothesis was rejected.

## 3. Results

### 3.1. Identification of the Species

With the help of the DNA barcoding method, based on the similarity between the sequences obtained in this study and sequences existing in the NCBI database, we identified the three species, VEUN20 as *V. mandarinia*, VENU21 as *V. basalis*, and VENU22 as *V. velutina*. The obtained sequences for the COI gene, corresponding to the “DNA Barcode” region, are available in GenBank under accession number MN477949- MN477951 (Appendix A).

### 3.2. Nutritional Composition of Vespa

#### 3.2.1. Amino Acids

Amino acid compositions of the *Vespa* species studied are represented in Table 1. There were 17 amino acids in all *Vespa* samples. There was a significant difference in the total amino acid content of broods of different species of *Vespa* (df = 3,4, F = 19.135, *p* = 0.008). Almost all the amino acids were found to be higher in the *V. velutina* brood collected from the wild in Korea. However, the differences were not significant in all cases (Table 1). Considering the totality of the amino acids as protein, the results show that *V. velutina* and *V. mandarinia* collected from the commercial production unit in Shizong province (China) had similar protein contents whereas *V. basalis* contained less protein. However, *V. velutina* broods collected from the forest near Andong (Korea) contained the highest amount of amino acids. Leucine was the predominating essential amino acid followed by lysine and valine. Tryptophan was not assessed and the amounts of cysteine and methionine were not measured in their entirety presumably because of the acid hydrolysis process [29]. Among the non-essential amino acids, glutamic acid was the most abundant one. The amino acid scoring pattern suggested that among all the estimated indispensable amino acids, methionine was found to be limiting; others satisfied (having a score of >100) the ideal protein pattern recommended by the WHO/FAO/UNU [26].

#### 3.2.2. Fatty Acids

Table 2 represents the fatty acid compositions of the *Vespa* broods studied. Significant differences were found in the total fatty acid content of broods of different *Vespa* species (df = 3,4, F = 12.255, *p* = 0.017). Palmitic acid was the predominating saturated fatty acid; it was followed by stearic acid. Oleic acid was the most abundant monounsaturated fatty acid. However, there was apparently no consistency in the polyunsaturated group. Except for *V. velutina* from China, linoleic acid was always found in abundance among the polyunsaturated fatty acids, although the quantities varied widely (0.55 to 9.49 mg/100 g). Overall, the *V. mandarinia* and *V. basalis* broods were found to have the highest proportions of polyunsaturated fatty acids. However, no such difference was found between the saturated and monounsaturated fatty acid contents in these two species. By contrast, the *V. velutina* brood contained a higher amount of saturated fatty acids followed by monounsaturated and polyunsaturated fatty acids.

#### 3.2.3. Mineral Content

Table 3 represents the comparative account of mineral contents of the species studied as well as the RDA values of respective elements. Significant differences in the mineral content, except manganese, of broods of different *Vespa* species were found (calcium: df = 3,4, F = 19.994, *p* = 0.007; magnesium: df = 3,4, F = 478.504, *p* = 0.000; sodium: df = 3,4, F = 161.653, *p* = 0.000; potassium: df = 3,4, F = 56.353, *p* = 0.001; phosphorus: df = 3,4, F = 42.778, *p* = 0.002; iron: df = 3,4, F= 38.232, *p* = 0.002; zinc: df = 3,4, F = 19.654, *p* = 0.007; manganese: df = 3,4, F = 4.086, *p* = 0.104; copper: df =3,4, F = 603.414, *p* = 0.000). The *V. velutina* brood collected from the wild in Korea was found to have a higher mineral content except for zinc and copper, however, the differences were not always significant. For magnesium, potassium, phosphorus, iron, and manganese, no significant differences were found between *V. velutina* broods from China and Korea. Potassium was the most abundant element. It was followed by phosphorus. Among the microminerals, iron and zinc were found to be dominating. The farmed *V. velutina* brood contained much less sodium than that detected in the wild-collected *V. velutina* brood. The potassium-to-sodium ratio (K/Na) in the wild *V. velutina* brood was as high as 11.7, but even higher ratios were noted in the farmed broods of *V. velutina*, *V. mandarinia*, and *V. basalis*, namely, 72.3, 13.7, and 45.5, respectively. It is noteworthy that the wild-collected *V. velutina* brood contained higher amounts of most of the minerals than the farmed hornets.

## 4. Discussion

### 4.1. Amino Acid Composition

The amino acid distributions seen in our study are in general agreement with a previously published report, although the contents of some of the amino acids in the current study were a little less than what was reported earlier [1]. The total amino acid content as protein content of *Vespa* broods is comparable with other reports on edible insects, including wasps (*V. velutina nigrithorax*: 48.64 [30], *V. mandarinia*: 59.7 [2], *V. basalis*: 43.91, *V. mandarinia*: 52.20, *V. velutina auraria*: 49.03, *V. tropica ducalis*: 42.44 [1]). In the earlier study, as with ours, overall glutamic acid was the most abundant amino acid. Glutamic acid is the precursor of GABA (gamma aminobutyric acid), which is a neurotransmitter of inhibitory neurons [31] and thus might be responsible for docile behavior. However, the saliva of *Vespa* spp. larvae is not rich in glutamic acid [32], which could be a reason for the wasps’ aggressiveness in defense and times of hunting prey. This aggressive behavior lessens in the nest where trophallaxis is performed, an essential behavioral trait for a social insect. The high proline content, amongst other effects, influences the flight of the insect, because it is metabolized to produce energy for the wing movements during flight [33]. It has been reported that hornet, with higher proline content in their saliva, generally build their nests higher up and also have a wider hunting range than hornets with less proline content in saliva and those live underground or nest in caves or tree holes and hunt over a smaller area [32].

Although *V. mandarinia* and *V. velutina* inhabit subterranean and open-air nests, respectively [34,35], they did not differ with respect to proline content, at least in the brood stage. Nonetheless, differences in body composition can be expected to exist between larvae, pupae, and adults as well as drones, workers, and queens with regard to developmental as well as physiological states, as has been demonstrated for the honeybee (*A. mellifera*) and the bumblebee (*Bombus terrestris*) [36,37,38,39]. Histidine, decarboxylated to histamine, is a major component of the venom and found in similar amounts in *V. velutina* and *V. mandarinia*, but less in the case of *V. basalis*. Among the indispensable amino acids, leucine was found to predominate and to be present in higher amounts in *V. velutina* and *V. mandarinia* than in *V. basalis*. Leucine and isoleucine are metabolized in the musculature.

From a nutritional standpoint, lysine deserves consideration as it is a limiting amino acid in cereals such as rice, wheat, and maize. Catabolism of lysine, an entirely ketogenic amino acid, includes the saccharopine pathway, which results in the formation of glutamate and α-aminoadipate. In addition, lysine is also a precursor for the biosynthesis of carnitine, which plays an important role in β-oxidation [40]. The aromatic amino acid tyrosine functions as precursor of catecholamines such as dopamine, norepinephrine, and epinephrine and is involved in melanogenesis. Tyrosine is a conditionally essential amino acid with phenylalanine playing a crucial role in tyrosine synthesis. Therefore, because of the presence of almost all proteinogenic amino acids and estimated indispensable ones except for methionine, satisfying the recommended protein pattern (Table 1), *Vespa* broods would supplement the nutritional requirements in people.

### 4.2. Fatty Acid Composition

The higher polyunsaturated acid contents of *V. mandarinia* and *V. basalis* are likely due to their diet, which consists of crickets and grasshoppers, which are often rich in polyunsaturated fatty acids (cf., grasshopper, *Chondacris rosea*: [41], and crickets, *Gryllus* sp. and *Teleogryllus* sp.: [42]). Compared to earlier analyses, the proportions of monounsaturated fatty acids were higher in the case of other hymenopteran species such as, to mention but a few, *B. ignitus* [37], *B. terrestris* [38], *Carebara vidua* [43], *Polyrhachis vicina* [44,45], and *Oecophylla smaragdina* [46]. However, diet manipulations can result in changes in the fatty acid composition in farmed versus wild insects [47,48]. 

Oleic acid was the dominant monounsaturated and palmitic acid, with stearic and myristic acids being the abundant saturated fatty acids. Linoleic acid was the most abundant polyunsaturated fatty acid followed by linolenic acid in *V. mandarinia* and *V. basalis*. Primarily saturated fatty acids such as myristic, palmitic, and lauric acids increased the level of low density lipoprotein, the so-called bad cholesterol. However, stearic acid does not raise serum cholesterol [49]. Oleic acid inhibits the store-operated Ca^+2^ entry process (SOCE) that controls the Ca^+2^ influx pathway and is involved in several cellular and physiological processes, including cell proliferations that are often diagnostic for colorectal cancer [50]. Oleic acid also shows significant in vitro inhibition of prolyl-endopeptidase (PEP), an enzyme playing a crucial role in the formation of amyloid in the brain and implicated in disorders such as dementia and Alzheimer’s disease [50].

Unsaturated fatty acids are being given special attention as they are seen to be beneficial to human health. Earlier studies, e.g., by Grundy [51], demonstrated the capacity of monounsaturated fatty acids to lower lipoprotein density and thus total cholesterol. Polyunsaturated fatty acids play a crucial role in the biosynthesis of cellular hormones such as eicosanoids and other signaling compounds that modulate human health [50]. Polyunsaturated fatty acids are of two types, i.e., *n*−6 and *n*−3, based on the position of the first unsaturated site in the chain. A higher ratio of *n*−3 to *n*−6, i.e., more *n*−3 and less *n*−6 in the diet, is preferable in connection with human health as it helps to reduce weight through the removal of intra-abdominal fat, a drop in adipocyte cell size, and the normalization of the heartbeat. Diets with high *n*−3 polyunsaturated fatty acids enhance the body’s ability to reduce damaging inflammatory conditions, as they are usually converted into anti-inflammatory eicosanoids. Besides, epidemiological as well as clinical studies further suggest that *n*−3 polyunsaturated fatty acids, including *n*−3 from marine food resources, help lower cardiovascular mortality [52,53] through mechanisms that include the modulation of cellular metabolic functions and gene expression and exert beneficial effects on lipid profiles and blood pressure [54]. Thus, the presence of linolenic acid in *V. mandarinia* and *V. basalis* could have nutritional benefits for sufferers of cardiovascular complexities.

### 4.3. Mineral Content

Minerals are essential micronutrients that play critical roles in human health. Among the minerals of nutritional importance, potassium was found in abundance followed by phosphorus. The values were within the range of reports on Hymenoptera as well as edible insects belonging to other orders [42,55]. A substantial amount of evidence shows that potassium intake lowers blood pressure. An increased intake of potassium also plays a critical role in the management of hypercalciuria and is likely to decrease the risk of osteoporosis [56]. Low serum potassium is strongly related to glucose intolerance and increases the risk of lethal ventricular arrhythmias in patients suffering from ischemic heart disease. On the other hand, a high intake of dietary sodium is associated with a prevalence of hypertension [57].

From the human nutritional point of view, a high potassium-to-sodium ratio can be regarded as beneficial as it reduces cardiovascular risk and improves blood pressure [58]. All the hornet broods contained higher potassium and comparatively less sodium, and thus can be regarded as beneficial for human health. The calcium content of the *Vespa* brood was found to be within the range of other insects [55], but less than what had been reported for *A. mellifera* larvae and pupae [36]. Over 99% of total body calcium by virtue of its phosphate salt is present in the teeth and bones of mammals and the remainder is present in the blood, extracellular fluid, muscle, and other tissues. Calcium plays critical physiological roles, including mediating vascular contraction and vasodilation, muscle contraction, nerve transmission, and glandular secretion, to mention a few [59]. Chronic calcium deficiency often results in a reduction in bone mass and osteoporosis [60], the development of hypertension, and even colon cancer [61].

Among the minerals of nutritional importance, iron receives the most attention. Iron deficiency and anemia still exist and are major public health concerns worldwide, especially in many developing and low-income countries [62,63]. The most vulnerable sections of the population in developing countries regarding iron deficiency are women of childbearing age and children under five [64,65]. Although the iron content of the studied species was a little less than that was reported for *A. mellifera* workers [36], it could still supplement the iron requirement, especially for those in the population who cannot afford iron-rich food such as red meat, liver, fish, etc. The element zinc plays a role in several cellular processes, including catalytic, structural, and regulatory roles in many enzymes, gene transcription, signal transduction pathways, etc. [66]. Although severe zinc deficiency is considered rare [67], mild to moderate zinc deficiencies persist worldwide [68]. An inadequate dietary zinc intake is particularly common in Sub-Saharan Africa and South Asia [69]. Based on the RDA values provided by agencies, consumption of 100 g of *Vespa* brood could satisfy a significant proportion of daily dietary requirements for minerals, especially iron, zinc, and copper (Table 3). Even if it cannot meet the total mineral need, assuming good bioavailability, the consumption of *Vespa* brood could at least supplement the nutritional requirements of minerals and could help mitigate the problems of mineral deficiencies in those sections of the population most at risk for them.

## 5. Conclusions

Since it was first suggested in 1975 that insects could help ease the problem of global food shortages [70], the last two decades in particular have seen a remarkable increase in attention to edible insects as a nutrient-rich and healthy food resource. Numerous countries have formulated or are in the process of formulating legislation to regulate mass production and trade of edible insects. The global market value of edible insects is expected to exceed USD 522 million by 2023 [71]. Since 2012, the South Korean edible-insect market focusing on human consumption alone (not including insects as animal feed) has experienced major advances, with governmental support and successful research endeavors [42,72]. The country’s tradition of using certain insects such as processed silkworm pupae, commonly known as *beondaegi*, as food [73] has further helped. Noting the competent nutrient composition of all three *Vespa* species examined by us and the feasibility of rearing these insects, we propose that these hornet broods can be a sustainable, high-quality nutritional source. However, the rearing process is yet to be established before any large-scale controlled production can commence.

## Figures and Tables

**Figure 1 foods-10-00418-f001:**
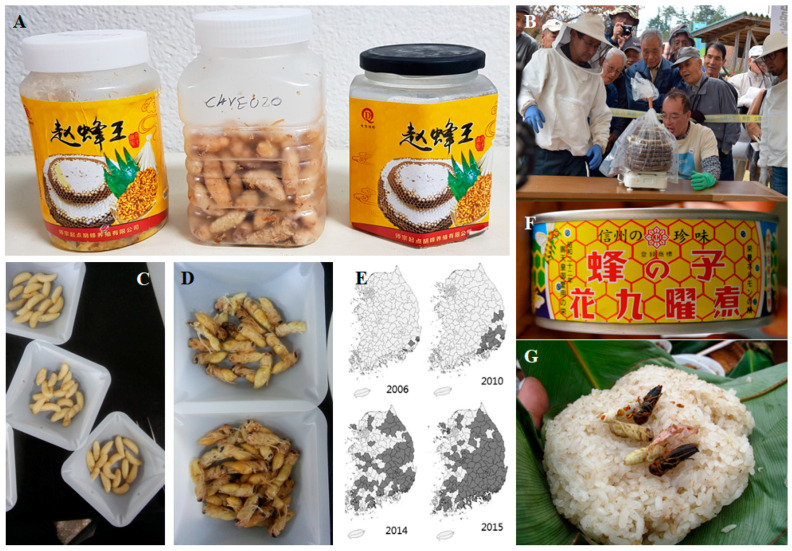
(**A**) bottled hornet larvae sold in China; (**B**) *hebo* (*Vespula* sp.) contest in which judges evaluate wasp nests by weight in search for the largest nest of domestically raised wasps. [Photo credit: Soleil Ho; source: https://www.splendidtable.org/story/the-japanese-tradition-of-raising-and-eating-wasps, accessed on 31 August 2019]; (**C**) larvae of *Vespa velutina* collected from the forest near Andong (Korea); (**D**) pupae of *Vespa velutina* collected from the forest near Andong (Korea); (**E**) range expansion of the invasive hornet *Vespa velutina* in Korea [Source: 19]; (**F**) canned *hachinoko* (wasp brood) in Japan; (**G**) practice of eating wasps in Yunnan province [Source: http://teaurchin.blogspot.com/2011/10/weird-things-ive-eaten-in-yunnan.html, accessed on 31 August 2019].

**Table 1 foods-10-00418-t001:** Amino acid composition (g/100 g dry matter, mean ± SD, duplicate analysis with composite sampling process) of different *Vespa* species broods, requirement of indispensable amino acid and amino acid scoring pattern as per the WHO/FAO/UNU [26] and amino acid scoring pattern (%).

	China			Korea	Indispensable Amino Acid Requirements ^1^	Amino Acid Scoring Pattern ^1^				
	*Vespa velutina*	*Vespa mandarinia*	*Vespa basalis*	*Vespa velutina*	mg/kg Per Day	mg/g Protein	Amino Acid Score			
							*Vespa velutina*	*Vespa mandarinia*	*Vespa basalis*	*Vespa velutina*
Leucine *	3.3 ± 0.18 ^b^	3.2 ± 0.33 ^b^	2.4 ± 0.16 ^c^	4.3 ± 0.32 ^a^	39	59	145.8	149.3	145.0	143.7
Valine *	2.3 ± 0.15 ^b^	2.3 ± 0.24 ^b^	1.6 ± 0.11 ^c^	3.2 ± 0.25 ^a^	26	39	152.2	158.3	149.9	161.5
Isoleucine *	2.1 ± 0.17 ^ab^	2.1 ± 0.19 ^b^	1.5 ± 0.07 ^c^	2.7 ± 0.28 ^a^	20	30	188.2	185.8	177.1	174.9
Methionine *	0.6 ± 0.04 ^a^	0.3 ± 0.33 ^a^	0.3 ± 0.07 ^a^	ND	10	16	92.3	57.8	73.5	ND
Cysteine	0.3 ± 0.28 ^a^	0.7 ± 0.68 ^a^	0.1 ± 0.01 ^a^	2.0 ± 1.13 ^a^	4	6	123.1	312.7	77.2	643.6
Lysine *	2.3 ± 0.03 ^ab^	2.3 ± 0.22 ^ab^	1.9 ± 0.23 ^b^	2.7 ± 0.07 ^a^	30	45	134.2	137.2	148.9	116.6
Threonine *	1.6 ± 0.05 ^b^	1.6 ± 0.12 ^b^	1.2 ± 0.13 ^b^	2.3 ± 0.14 ^a^	15	23	188.1	183.2	182.9	193.7
Histidine *	1.2 ± 0.09 ^b^	1.2 ± 0.21 ^b^	0.9 ± 0.06 ^b^	1.7 ± 0.14 ^a^	10	15	212.8	208.5	211.5	217.8
Phenylalanine *	1.6 ± 0.01 ^ab^	1.6 ± 0.39 ^ab^	1.2 ± 0.16 ^b^	2.1 ± 0.07 ^a^	25	38	286.7	311.2	296.5	283.0
Tyrosine **	2.5 ± 0.10 ^ab^	2.7 ± 0.38 ^ab^	2.0 ± 0.13 ^b^	3.3 ± 0.46 ^a^						
Arginine ***	1.7 ± 0.05 ^ab^	0.8 ± 0.86 ^b^	1.2 ± 0.13 ^ab^	2.2 ± 0.18 ^a^						
Aspartic acid	2.4 ± 0.06 ^b^	2.4 ± 0.45 ^b^	1.8 ± 0.28 ^b^	3.7 ± 0.04 ^a^						
Glutamic acid	7.6 ± 0.19 ^a^	7.8 ± 1.6 ^a^	6.2 ± 1.04 ^a^	9.0 ± 0.96 ^a^						
Serine	1.7 ± 0.12 ^b^	1.6 ± 0.09 ^bc^	1.2 ± 0.13 ^c^	2.4 ± 0.28 ^a^						
Proline	2.3 ± 0.23 ^a^	2.1 ± 0.17 ^a^	1.6 ± 0.05 ^b^	2.4 ± 0.11 ^a^						
Glycine	2.4 ± 0.60 ^ab^	2.3 ± 0.34 ^ab^	1.6 ± 0.13 ^b^	3.4 ± 1.03 ^a^						
Alanine	2.1 ± 0.69 ^a^	2.0 ± 0.51 ^a^	1.4 ± 0.16 ^a^	3.4 ± 1.27 ^a^						
**Total**	**37.9 ± 1.65 ^b^**	**36.8 ± 2.37 ^b^**	**28.1 ± 2.22 ^c^**	**50.5 ± 4.74 ^a^**						

* Essential amino acid. ** Conditional essential amino acid. *** Essential amino acid for children. ^1^ All the values were obtained for the adult population from the WHO/FAO/UNU 2007 report of a joint WHO/FAO/UNU Expert Consultation on protein and amino acid requirements in human nutrition. ND = Not detected. Superscripts indicate significant difference (*p* ≤ 0.05).

**Table 2 foods-10-00418-t002:** Fatty acid composition (g/100 g dry matter, mean ± SD, duplicate analysis with composite sampling process) of different *Vespa* species broods. Superscripts indicate a significant difference (*p* ≤ 0.05) for selected predominating fatty acids.

	China			Korea
	*Vespa velutina*	*Vespa mandarinia*	*Vespa basalis*	*Vespa velutina*
**Saturated Fatty Acids**				
Capric acid	ND	ND	ND	<0.01
Lauric acid	0.2 ± 0.04	0.2 ± 0.04	0.1 ± 0.02	0.3 ± 0.02
Tridecanoic acid	ND	ND	ND	0.01 ± 0.00
Myristic acid	0.7 ± 0.14 ^a^	0.5 ± 0.17 ^ab^	0.3 ± 0.06 ^b^	0.8 ± 0.11 ^a^
Palmitic acid	3.7 ± 0.49 ^a^	4.3 ± 0.24 ^a^	3.5 ± 0.01 ^a^	3.5 ± 0.28 ^a^
Heptadecanoic acid	0.02 ± 0.00	0.02 ± 0.00	0.04 ± 0.00	0.02 ± 0.00
Stearic acid	0.9 ± 0.06	1.0 ± 0.17	1.2 ± 0.01	0.7 ± 0.00
Arachidic acid	0.1 ± 0.01	0.2 ± 0.03	0.2 ± 0.02	0.1 ± 0.00
Behenic acid	ND	0.1 ± 0.02	0.1 ± 0.01	0.1 ± 0.00
Lignoceric acid	ND	0.01 ± 0.01	0.03 ± 0.00	0.1 ± 0.01
**Subtotal**	**5.6 ± 0.71 ^a^**	**6.2 ± 0.21 ^a^**	**5.4 ± 0.05 ^a^**	**5.4 ± 0.42 ^a^**
**Monounsaturated Fatty Acids**				
Myristoleic acid	0.02 ± 0.01	ND	ND	0.03 ± 0.01
Palmitoleic acid	0.4 ± 0.10	0.2 ± 0.05	0.1 ± 0.00	0.4 ± 0.06
cis-10-Heptadecenoic acid	0.01 ± 0.02	ND	ND	0.01 ± 0.00
Oleic acid	4.1 ± 0.44 ^b^	5.6 ± 0.55 ^a^	5.3 ± 0.29 ^a^	4.1 ± 0.28 ^b^
cis-11-Eocosenic acid	ND	0.1 ± 0.06	0.14 ± 0.027	0.5 ± 0.01
**Subtotal**	**4.6 ± 0.56 ^a^**	**5.9 ± 0.56 ^a^**	**5.6 ± 0.31 ^a^**	**5.1 ± 0.37 ^a^**
**Polyunsaturated Fatty Acids**				
Linoleic acid	0.6 ± 0.11 ^b^	6.8 ± 3.09 ^a^	9.5 ± 1.96 ^a^	0.6 ± 0.02 ^b^
Linolenic acid	0.8 ± 0.11	1.2 ± 0.30	1.8 ± 0.01	ND
Arachidonic acid	0.04 ± 0.03	ND	ND	0.02 ± 0.00
cis-5,8,11,14,17-Eicosapentaenoic acid	0.03 ± 0.02	ND	ND	ND
**Subtotal**	**1.4 ± 0.05 ^b^**	**8.1 ± 3.39 ^a^**	**11.2 ± 1.98 ^a^**	**0.6 ± 0.04 ^b^**
**Total**	**11.5 ± 1.31 ^b^**	**20.1 ± 3.75 ^a^**	**22.2 ± 2.25 ^a^**	**11.1 ± 0.82 ^b^**

ND = Not detected. Superscripts indicate significant difference (*p* ≤ 0.05).

**Table 3 foods-10-00418-t003:** Mineral contents (mg/100 g dry matter, mean ± SD, duplicate analysis with composite sampling process) of different *Vespa* species broods and recommended dietary allowance (RDA) or population reference intakes (PRI) and satisfying the requirement in %

	China			Korea	RDA ^1^		PRI/AI ^2^		Satisfying the Requirement as per PRI/AI^2^ by 100 g of Consumption of Respective *Vespa* Brood (in %) ^3^							
									*Vespa velutina*		*Vespa mandarinia*		*Vespa basalis*		*Vespa velutina*	
	*Vespa velutina*	*Vespa mandarinia*	*Vespa basalis*	*Vespa velutina*	M	F	M	F	M	F	M	F	M	F	M	F
**Ca**	38.8 ± 0.04 ^b^	27.4 ± 0.20 ^c^	31.8 ± 0.34 ^bc^	46.3 ± 5.22 ^a^	1000		950		4.1		2.9		3.3		4.9	
**Mg**	63.9 ± 0.01 ^a^	33.0 ± 0.44 ^c^	38.2 ± 0.24 ^b^	66.3 ± 2.16 ^a^	400	310	350 ^†^	300 ^†^	18.3	21.3	9.4	11.0	10.9	12.7	18.9	22.1
**Na**	10.4 ± 0.02 ^c^	30.8 ± 0.40 ^b^	8.9 ± 0.09 ^c^	61.5 ± 5.44 ^a^	1500 *	3800 *	--	--	0.7	0.3	2.1	0.8	0.6	0.2	4.1	1.6
**K**	751.6 ± 0.87 ^a^	422.7 ± 6.58 ^b^	404.4 ± 0.01 ^b^	718.6 ± 69.87 ^a^	3400 *	2600 *	3500 ^†^		21.5		12.1		11.6		20.5	
**P**	561.2 ± 1.18 ^a^	322.5 ± 2.93 ^b^	318.4 ± 4.90 ^b^	641.9 ± 71.37 ^a^	700		550 ^†^		102.0		58.6		57.9		116.7	
**Fe**	10.0 ± 0.12 ^a^	7.2 ± 0.41 ^b^	5.0 ± 0.18 ^c^	9.1 ± 0.89 ^a^	8	18	11	16	90.9	62.5	65.5	45.0	45.5	31.3	82.7	56.9
**Zn**	7.2 ± 0.02 ^a^	4.7 ± 0.01 ^c^	5.1 ± 0.04 ^c^	6.1 ± 0.71 ^b^	11	8	9.4	7.5	76.6	96.0	50.0	62.7	54.3	68.0	64.9	81.3
**Mn**	0.6 ± 0.02 ^ab^	0.1 ± 0.01 ^b^	1.2 ± 0.68 ^ab^	2.8 ± 1.49 ^a^	2.3 *	1.8 *	3 ^†^		20.0		3.3		40.0		93.3	
**Cu**	2.2 ± 0.04 ^a^	0.9 ± 0.01 ^d^	1.1 ± 0.04 ^c^	1.3 ± 0.04 ^b^	900		1.6 ^†^	1.3 ^†^	137.5	169.2	56.3	69.2	68.8	84.6	81.3	100.0

^1^ Recommended dietary allowance (RDA) values were obtained from the Micronutrient Information Center, Linus Pauling Institute, Oregon State University [www.lpi.oregonstate.edu/mic/minerals, accessed 9 February 2021]. All the values are for adult (>19–50) populations and provided in mg/day except for copper, which is in µg/day. M = male; F = female. * indicates the adequate intake (AI) values. ^2^ Population reference intakes (PRIs) and adequate intakes (AIs) for minerals were obtained from the European Food Safety Authority (EFSA) [www.efsa.europa.eu/sites/deafult/files/assets/DRV_Summary_tables_jan_17.pdf, accessed 9 February 2021]. All values are for adult (>25 years) populations and provided in mg/day. ^†^ indicates the adequate intake (AI) values. ^3^ Sodium (Na) was calculated based on the value provided by the Linus Pauling Institute, Oregon State University; all others were based on values provided by the European Food Safety Authority (EFSA). Superscripts indicate significant difference (*p* ≤ 0.05).

## Data Availability

Not applicable.

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
