# Peer review of "Chemical Composition and Nutritional Value of Different Species of Vespa Hornets"

_foods, 2021, doi:10.3390/foods10020418_

Round 1
Reviewer 1 Report
I haven't read such an understandable, concise, and well-written article for a long time, which I highly praise. I also appreciate numerous references.
I have only a few suggestions:
In all tables and in the whole text the authors could consider reducing the decimal places to 2 (now somewhere there are 3, somewhere 2, and I am not sure there is a reason for the difference).
I also suggest adding another column with the value of DDD (recommended daily dose according to WHO tables) to the tables of nutritional values ​​- so that it is clear how many of those larvae and pupae need to be eaten, so that it has an effect or, there can be a comparison with common foods - meat, oil...
Just something to think about: The authors mention the possibility of mass rearing – a source of suitable feed must be considered if this should be realized. Because, with improper feed, the nutritional quality of the insect bodies might be very different (probably worse).
Author Response
Reviewer 1
I haven't read such an understandable, concise, and well-written article for a long time, which I highly praise. I also appreciate numerous references.
I have only a few suggestions:
In all tables and in the whole text the authors could consider reducing the decimal places to 2 (now somewhere there are 3, somewhere 2, and I am not sure there is a reason for the difference).
Thank you for the comments. We have reduced decimal places to 1 for the mean and decimal places 2 for the SD for the amino acid and minerals. As the values are small we keep the fatty acid table as it is.
I also suggest adding another column with the value of DDD (recommended daily dose according to WHO tables) to the tables of nutritional values ​​- so that it is clear how many of those larvae and pupae need to be eaten, so that it has an effect or, there can be a comparison with common foods - meat, oil...
We understand. We have now discussed the importance of the nutrient content in the light of requirements. We also calculated the amino acid score based on FAO/WHO/UNU consultation protein and amino acids requirement values and discussed. Also, we discussed RDA values for mineral requirement and included in the respective table.
Just something to think about: The authors mention the possibility of mass rearing – a source of suitable feed must be considered if this should be realized. Because, with improper feed, the nutritional quality of the insect bodies might be very different (probably worse).
Yes, we agree that feed is really important for the mass production. We are working on the Vespa rearing system for the large production (domestication/ semi-domestication).
Reviewer 2 Report
Dear Authors,
your paper has been sent for my consideration. In it, you investigate the nutritional value and chemical composition different species of Vespa. The manuscript was prepared very carefully. The introduction to the topic is correct and sufficient. The applied analytical methods were selected correctly and the results were presented and discussed in a clear and legible way.
I just have a few minor comments:
- 2.3.1. Please calculate and add the amino acid score (AAS) for adults using the standard method recommended by the FAO
- 2.3.3.: The percent of population reference intake (PRI) and adequate intake (AI) according to the latest EFSA recommendations should be also calculated and added to Table 3.
- A discussion of the above data should be added to the manuscript.
- In the Discussion chapter, incorrect numbering of subsections was used.
Author Response
Reviewer 2
Dear Authors,
your paper has been sent for my consideration. In it, you investigate the nutritional value and chemical composition different species of Vespa. The manuscript was prepared very carefully. The introduction to the topic is correct and sufficient. The applied analytical methods were selected correctly and the results were presented and discussed in a clear and legible way.
I just have a few minor comments:
2.3.1. Please calculate and add the amino acid score (AAS) for adults using the standard method recommended by the FAO
==> Really thanks for the comments. Yes, now we have calculated the amino acid score and included in the table and also discussed.
2.3.3.: The percent of population reference intake (PRI) and adequate intake (AI) according to the latest EFSA recommendations should be also calculated and added to Table 3.
==> Yes, now we have included the PRI and AI values recommended by EFSA in the Table and discussed it. Also we have calculated the % satisfying by consuming 100g of brood.
A discussion of the above data should be added to the manuscript.
==> We have discussed now.
In the Discussion chapter, incorrect numbering of subsections was used.
==> Thank you. We have corrected.
Reviewer 3 Report
In the paper “Chemical composition and nutritional value of different species of Vespa hornets”, the authors Sampat Ghosh, Chuleui Jung, Saeed Mahamadzade Namin and Victor Benno Meyer-Rochow compare the composition and nutritional values of larvae and pupae of 3 species of hornet obtained from a Chinese breeding facility in Shizong Province (V. velutina, V. mandarinia and V. basalis) and a wild hornet species (V. velutina) whose nest was collected in South Korea in the city of Handong. The study compares the amino acid, fatty acid and mineral contents of pupae and larvae. The few results obtained in this study are very interesting and the authors, thanks to a well-documented discussion, demonstrate the value of raising hornets for human consumption. This study is entirely in line with the global context of population growth, which requires better characterization and use of food resources, particularly proteins.
Nevertheless, some points must be clarified.
Key Notes:
The authors need to better present the sampling strategy. If it is clear that the adults were excluded from the study, the proportions of larvae and pupae are not mentioned in the “material and methods” section. Indeed, it is not possible to know if these proportions are similar for the 4 broods. In addition, it is not mentioned for nests from China how many individuals were used while this number (n=100) is indicated for wild brood. This information is fundamental and necessary for the interpretation of the results because the amino acid, mineral and fatty acid compositions vary during the development of the insect. What the authors also pointed out (l 236 -239).
The authors shall indicate in the legends of tables 1, 2 and 3 the number of pupae and larvae that have been used and shall also specify the number of technical replicates used for the mean and SD calculations. In addition, for consistency and accuracy, the authors should perform a post-hoc test following ANOVA to indicate significant differences between brood (for amino acids, fatty acids and minerals).
Another important point concerns the precision of species comparisons in the discussion. Indeed, the values (amino acid concentrations, fatty acids, etc.) of the other studies are never cited, as are the experimental conditions (larvae, larvae + pupae, adults, etc.) of the previous studies cited in this work. It is therefore difficult for the reader to judge the relevance of the comparisons made by authors. The authors must provide this clarification.
Author Response
Reviewer 3
In the paper “Chemical composition and nutritional value of different species of Vespa hornets”, the authors Sampat Ghosh, Chuleui Jung, Saeed Mahamadzade Namin and Victor Benno Meyer-Rochow compare the composition and nutritional values of larvae and pupae of 3 species of hornet obtained from a Chinese breeding facility in Shizong Province (V. velutina, V. mandarinia and V. basalis) and a wild hornet species (V. velutina) whose nest was collected in South Korea in the city of Handong. The study compares the amino acid, fatty acid and mineral contents of pupae and larvae. The few results obtained in this study are very interesting and the authors, thanks to a well-documented discussion, demonstrate the value of raising hornets for human consumption. This study is entirely in line with the global context of population growth, which requires better characterization and use of food resources, particularly proteins.
Nevertheless, some points must be clarified.
Key Notes:
The authors need to better present the sampling strategy. If it is clear that the adults were excluded from the study, the proportions of larvae and pupae are not mentioned in the “material and methods” section. Indeed, it is not possible to know if these proportions are similar for the 4 broods. In addition, it is not mentioned for nests from China how many individuals were used while this number (n=100) is indicated for wild brood. This information is fundamental and necessary for the interpretation of the results because the amino acid, mineral and fatty acid compositions vary during the development of the insect. What the authors also pointed out (l 236 -239).
=> Thank you for the suggestion. We have now mentioned that the bottle comprises of late instar larvae and pupae of early stages. The proportion of both the stages was 50:50. Therefore we also separated the individuals from our harvested nest in Korea similar development stage and take equal no. of larvae and pupae for the analysis.
The authors shall indicate in the legends of tables 1, 2 and 3 the number of pupae and larvae that have been used and shall also specify the number of technical replicates used for the mean and SD calculations. In addition, for consistency and accuracy, the authors should perform a post-hoc test following ANOVA to indicate significant differences between brood (for amino acids, fatty acids and minerals).
=> Thank you for the advice. We carried out post hoc test i.e. Tukey HSD. Now we have included in the tables.
Another important point concerns the precision of species comparisons in the discussion. Indeed, the values (amino acid concentrations, fatty acids, etc.) of the other studies are never cited, as are the experimental conditions (larvae, larvae + pupae, adults, etc.) of the previous studies cited in this work. It is therefore difficult for the reader to judge the relevance of the comparisons made by authors. The authors must provide this clarification.
=> Now we have discussed with the values reported such as follows. Considering the total amino acids content as protein content of Vespa broods are comparable with other reports on edible insects including wasps (V. velutina nigrithorax: 48.64 [30]; V. mandarinia: 59.7 [2]; V. basalis: 43.91; V. mandarinia: 52.20; V. velutina auraria: 49.03; V. tropica duealis: 42.44 [1])..